# Identifying SNP markers associated with distinctness, uniformity, and stability testing in Egyptian fenugreek genotypes

**Mustafa M. H. Abd El-Wahab**[1], **Hashim Abdel-Lattif**[1], **Kh. S. Emara**[2], **Mohamed Mosalam**[3], **Maha Aljabri**[4], **Mohamed El-Soda**[5]*

1 Department of Agronomy, Faculty of Agriculture, Cairo University, Giza, Egypt, 2 Department of Agricultural Botany, Faculty of Agriculture, Cairo University, Giza, Egypt, 3 Department of Biotechnology, Faculty of Agriculture, Heliopolis University, Cairo, Egypt, 4 Department of Biology, Faculty of Applied Sciences, Umm Al-Qura University, Makkah, Saudi Arabia, 5 Department of Genetics, Faculty of Agriculture, Cairo University, Giza, Egypt

* mohamed.elsoda@agr.cu.edu.eg

**Data Availability Statement:** All relevant data are within the paper and its Supporting Information files.

## Abstract

Distinctness, uniformity, and stability (DUS) test is the legal requirement in crop breeding to grant the intellectual property right for new varieties by evaluating their morphological characteristics across environments. On the other hand, molecular markers accurately identify genetic variations and validate the purity of the cultivars. Therefore, genomic DUS can improve the efficiency of traditional DUS testing. In this study, 112 Egyptian fenugreek genotypes were grown in Egypt at two locations: Wadi El-Natrun (Wadi), El-Beheira Governorate, with salty and sandy soil, and Giza, Giza governorate, with loamy clay soil. Twelve traits were measured, of which four showed a high correlation above 0.94 over the two locations. We observed significant genotype-by-location interactions (GxL) for seed yield, as it was superior in Wadi, with few overlapping genotypes with Giza. We attribute this superiority in Wadi to the maternal habitat, as most genotypes grew in governorates with newly reclaimed salty and sandy soil. As a first step toward genomic DUS, we performed an association study, and out of 38,142 SNPs, we identified 39 SNPs demonstrating conditional neutrality and four showing pleiotropic effects. Forty additional SNPs overlapped between both locations, each showing a similar impact on the associated trait. Our findings highlight the importance of GxL in validating the effect of each SNP to make better decisions about its suitability in the marker-assisted breeding program and demonstrate its potential use in registering new plant varieties.

## Introduction

The genus *Trigonella* belongs to the Fabaceae family. It comprises about 260 species, of which three are the most popular by the vernacular name fenugreek: *T. foenum-graceum* (fenugreek), *T. hamosa* (Egyptian fenugreek), and *T. laciniata* (Jagged fenugreek) [1, 2]. The diploid fenugreek species is an annual legume crop widely cultivated in North and East Africa, South East

**Funding:** The author(s) received no specific funding for this work.

**Competing interests:** The authors have declared that no competing interests exist.

Asia, Mediterranean and Eastern Europe, the U.S.A., Canada, and Argentina. Fenugreek can be used either as fresh leaves, chopped leaves (flavouring agent), sprouts (salad), pot herbs (decoration), seed extracts, or powders (spice, condiments, or medicines) [3–6]. The description and benefits of fenugreek had been reported as early as 1500 BC in the Egyptian Ebers Papyrus, one of the oldest maintained medicinal documents [7, 8].

Fenugreek grows well under a wide range of conditions; it is moderately tolerant to drought and salinity and can even be profitably grown on marginal lands such as sandy soil [8, 9]. In addition, fenugreek, as a legume, may fix about 283 kg N ha$^{-1}$ year$^{-1}$, improving marginal"-lands' fertility [10]. Fenugreek resembles a large clover and can be used as a forage [8] with a long cylindrical (30–60 cm long) and pinkish stem. Roots are massive finger-like structures [11]. Fenugreek has pinnate, trifoliate, and long-stalked compound leaves. Leaves can be either toothed, lanceolate, stipules triangular, or obovate to oblanceolate leaflets [12]. The flower produces brownish to yellowish-brown ∼15 cm long 2–8 pods, each containing 10–20 seeds, ∼5 mm long, hard, smooth, dull yellow to brownish-yellow [13, 14].

The International Union for the Protection of New Varieties of Plants (UPOV) established plant variety rights (PVR) as an intellectual property form to safeguard"breeders' investments in developing new varieties, fostering innovation, and meeting consumer demand. To receive PVR protection, a new plant variety must pass a distinctness, uniformity, and stability (DUS) test, which requires the new variety to be distinct, stand out from well-known types, uniform in terms of the seeds that comprise the variety, and stable in various environments [15]. This test collects morphological characteristics influenced by developmental stages and environmental factors [16]. On the other hand, molecular markers provide more accurate information, are stable across developmental stages and environmental factors, and are distributed throughout the genome; thus, they are widely used in genetic diversity analysis [17].

Identical phenotypes can be expressed from different genetic backgrounds; thus, molecular markers can overcome the limitations of the DUS test and can be effectively used for variety identification. Therefore, the UPOV approved the application of molecular markers on the DUS test under three models. The first model is based on a close link between molecular markers and a specific gene. The second case investigates the correlations between molecular and phenotypic distances of varieties. The third model determines the distinctness between varieties based on at least three different SNPs before carrying out the uniformity and stability field trials [18].

Such DUS traits are quantitative traits controlled by many genes and influenced by the environment [19]. The genomic regions harbouring those genes are called quantitative trait loci (QTL) and can be mapped via the classical QTL approach using bi-parental populations. However, this approach suffers from the limited variation in the mapping population and the low mapping resolution. Therefore, the linkage disequilibrium (LD) based mapping approach, also known as association mapping, was introduced. This approach depends on phenotyping many genotypes collected from naturally evolved and adapted populations with wider genetic variation. Association mapping can often identify smaller intervals because of the historical recombination events over thousands of generations. Such recombination events can be detected using polymorphic markers such as single nucleotide polymorphisms (SNPs). These associations and candidate genes may provide key markers for trait introgression, marker-assisted selection, or targets for functional manipulation for crop improvement [17].

We planned this work to 1) evaluate the morphological and yield traits of different fenugreek genotypes collected from different places in Egypt under saline sandy and regular clay loamy soils, 2) evaluate the examined traits for the DUS test, 3) identify the significant SNPs associated with the examined traits, and select the superior genotypes, and 4) assess the application potential of SNP markers in assisting the DUS testing of fenugreek genotypes.

## Materials and methods

### Experimental site

Two field experiments were conducted during the growing season of 2017/2018 at two locations. The first location was the Desert Experimental Station, Faculty of Agriculture, Cairo University, Wadi El-Natrun, El-Beheira Governorate, Egypt, hereafter called Wadi. The second location was the Agricultural Research and Experiment Station, Faculty of Agriculture, Cairo University, Giza, Egypt, hereafter called Giza. The irrigation systems at Wadi and Giza are drip and flooding, respectively.

Climatic data of the Wadi and Giza experimental locations were as follows: the average temperature in October was 22.1 and 26.4˚C, then in January was 14.7 and 16.3˚C, and finally, in April was 25.4 and 29.3˚C, respectively. The highest relative humidity values were 58.6 in February and 54.5% in January at Wadi and Giza, respectively. Soil physical and chemical analysis of the two locations is presented in Table 1.

### Experimental design and treatments

We used 112 fenugreek genotypes collected directly from local farmers in several Egyptian governorates. For example, seven, 20, 17, and 15 genotypes were collected from Qalyubia, Qena, Beni-Suef, and Minya governorates, Egypt [20]. We used a complete randomized block design with three replications. Each plot consisted of 4 ridges of 0.70 m in width and 5.0 m in length. The experimental plot area was 14.0 $m^2$, with 200 plants/pot.

### Fieldwork procedures

The preceding crop was peanut (*Arachis hypogaea* L.) and corn (*Zea mays* L.) in Wadi and Giza, respectively. Sowing dates were October 12th and 7th, 2017, respectively. Seeds were sown in hills 20 cm apart by hand on both sides of the ridge. The applied nitrogen was ammonium nitrate (33.5% N), added once as 37.5 kg per ha at the sowing date. Calcium superphosphate fertilizer was added uniformly before sowing at 50 kg $P_2O_5$ /ha (15.5% $P_2O_5$). The weed management was carried out twice during the growing season by hoeing.

**Table 1. Physical and chemical properties of the upper 30 cm soil at the experimental site of both locations.**

| Character | Locations | |
|---|---|---|
| | **Wadi** | **Giza** |
| **Physical analysis** | | |
| Sand (%) | 93.2 | 28 |
| Silt (%) | 4.7 | 33 |
| Clay (%) | 2.1 | 39 |
| **Soil type** | Sandy | Clay loam |
| **Chemical analysis** | | |
| Available N (mg/kg) | 7.3 | 10.4 |
| Available P (mg/kg) | 5.8 | 15.2 |
| Available K (mg/kg) | 77.9 | 54.7 |
| Organic matter (%) | 1.67 | 2.43 |
| pH | 7.44 | 7.3 |
| EC (m/mohs/cm) | 5.37 | 0.88 |
| **Irrigation system** | Drip | Flooding |

## Phenotypic data collection

**A. Distinctiveness, uniformity, and stability (DUS).** Before harvesting, we excluded guarded plants and randomly chose ten plants from each plot to measure the following DUS traits according to the International Union for the Protection of New Varieties of Plants (UPOV 2013) and Protection of Plant Varieties and"Farmers' Rights Authority (PPVFRA 2016) with some modifications:

1. Cotyledonary leaf: recorded ten days after sowing as follows;

    a. Apex shape (acute or rounded),

    b. Basal shape (acute, obtuse, or round),

2. Prophyll: scored 20 days after planting as follows;

    a. Apex shape (acute or round),

    b. Basal shape (acute, obtuse, or round),

    c. Size based on length and width (short and thin, short and wide, tall and thin, and tall and wide).

3. The first leaf on the first branch: scored 38 days after sowing as follows;

    a. Apex shape (acute or rounded),

    b. Basal shape (acute, obtuse, or round),

    c. Size based on length and width (short and thin, short and wide, tall and thin, and tall and wide).

4. Stem: measured 120 days after sowing

    a. Plant height was scored either short, if <45 cm, or tall if > 45cm,

    b. The number of primary branches was scored low if <6 or high if >6.

5. Pod: measured 135 days after sowing

    a. Curvature (moderately curved or strongly curved).

6. Seed yield (SY): At harvesting time, 135 days from sowing, we measured SY/plot ($1m^2$).

## Statistical analysis

The raw data of the DUS traits of each experiment were statistically analyzed using SPSS v21. For SY, analysis of variance (ANOVA) tested the significant differences between genotypes (G), locations (L), and their interaction (G×L). Broad-sense heritability ($H^2$) was estimated as the ratio between the genetic variance ($\delta^2 g$) and the total phenotypic variance ($\delta^2 ph$) as follows;

$$H^2 = \delta^2 g / \delta^2 ph$$

$$\delta^2 ph = \delta^2 g + \delta^2 e$$

$$\delta^2 g = (MS_g - MS_e)/r$$

$$\delta^2 e = MS_e$$

Where $\delta^2 e$ is the error variance, $MS_g$ is the mean square of genotypes, $MS_e$ is the error mean square, and r is the number of replications.

The Pearson correlation was performed in R for Windows 4.1.2, using the R package corrplot.

## SNP data and association analysis

The 112 fenugreek genotypes were genotyped using the double digest restriction site-associated DNA sequencing (ddRAD-seq) technique [21]. Briefly, 38,142 bi-allelic SNPs with a minimum quality score of 30, a maximum of 0.2% missing data per SNP, a minimum mean coverage depth of 20, and a minimum minor allele frequency (MAF) of 0.05 were retained for association analysis [20].

The TASSEL software, version 5.0 [22], assisted in identifying SNP markers associated with the raw data of the measured traits using the mixed linear model (MLM), kinship matrix, and principal component. Based on Bonferroni correction and false discovery rate (FDR) of $\alpha =$ 0.05, the threshold was 6.5 and 5.9, respectively. We also report SNPs with thresholds above three as potential SNP, primarily if the SNP is associated with the same trait over the two locations.

## Results

### Statistical analysis

Based on the apex shape of the cotyledonary leaf, 42 and 65 genotypes were acute and round, respectively. Comparing the basal shape of the cotyledonary leaf in Wadi and Giza revealed 67 and 67 acute genotypes, 42 and 39 obtuse genotypes, and 3 and 6 round genotypes, respectively. Regarding the prophyll, we observed 32 and 26 acute genotypes and 80 and 86 obtuse genotypes in Wadi and Giza, respectively. The first leaf on the first branch was acute for 46 and 41 genotypes or obtuse for 66 and 71 in Wadi and Giza, respectively. The number of genotypes per each prophyll size category was 18 and 24 tall and wide, 40 and 42 short and wide, 13 and 21 tall and thin, and 41 and 25 short and thin, at Wadi and Giza, respectively. The first leaf on the first branch was tall and wide for 29 and 16 genotypes, short and wide for 46 and 42 genotypes, tall and thin for 19 and 0 genotypes, and short and thin for 18 and 53 genotypes in Wadi and Giza, respectively.

Plant height showed an apparent discrepancy between Wadi and Giza, where 32 and 55 genotypes were short, and 80 and 57 were tall, respectively. In contrast, this discrepancy was less pronounced in the case of the number of primary branches, where 32 and 42 genotypes had less than six branches, and 80 and 70 had more than six branches in Wadi and Giza, respectively.

In the case of pod curvature, no differences were observed between both locations, where 49 genotypes were moderately curved, whereas 63 genotypes were strongly curved.

The maximum SY was more remarkable for genotypes grown at Wadi than in Giza. For example, the four genotypes, G15, G98, G24, and G109, showed superior SY, 409, 407.7, 330.8, and 330.4 g/p, respectively. Comparing SY at both locations revealed few overlapping genotypes. Heritabilities were 0.98 and 0.92 at Wadi and Giza, respectively. ANOVA showed significant differences between genotypes and between locations, where the averages were 148.9 g and 102.2 g at Wadi and Giza, respectively. We observed significant genotype by location interaction (G x L).

Correlation analysis (Fig 1) showed significant positive correlations between the apex and basal shapes of the first leaf at the two locations. The same trend was observed between plant height and the number of primary branches. A negative correlation was observed between the

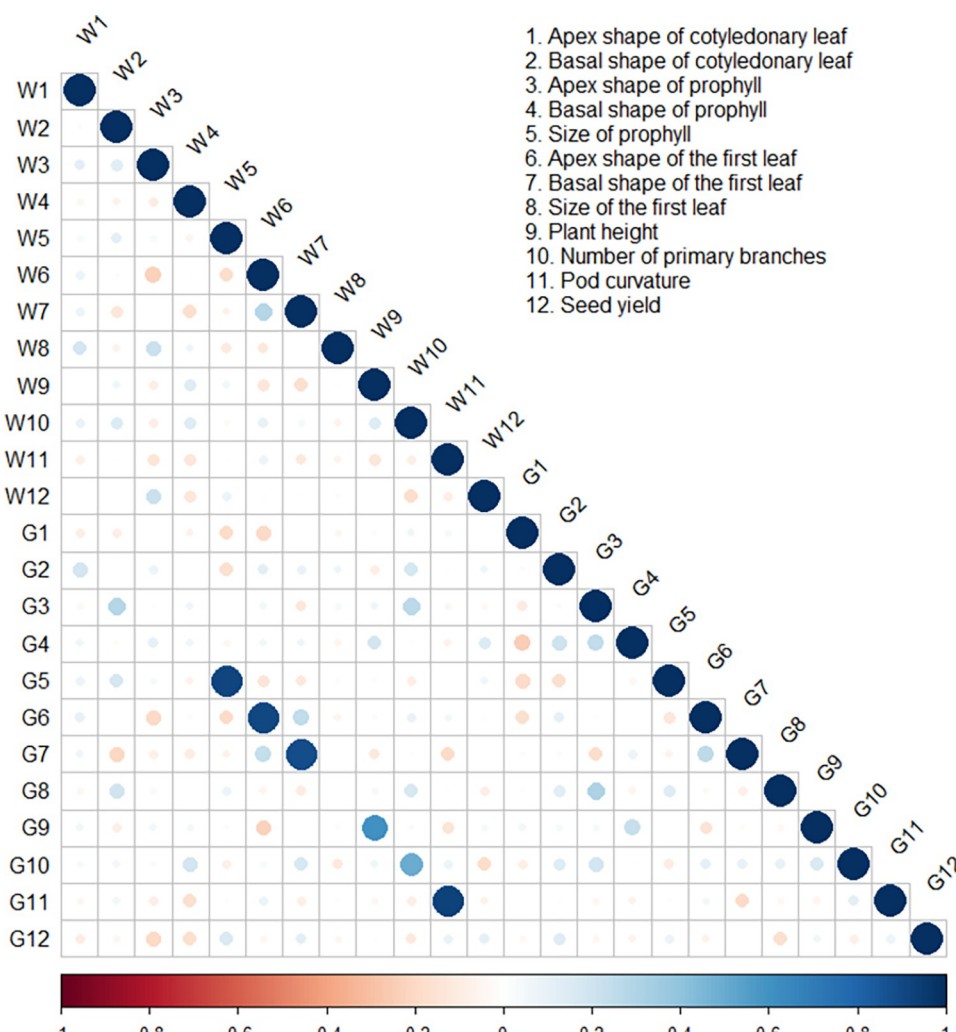

**Fig 1. Correlation plot between the measured traits at Wadi (W) and Giza (G).** Numbers indicate the measured traits as shown in the legend. The size and the colour of the circles indicate the significance level. The blue and red colours indicate positive and negative correlations, respectively.

apex shape of the first leaf and plant height at the two locations. Comparing the same trait over the two locations revealed positive correlations in the case of the size of the prophyll, the apex shape of the first leaf, the basal shape of the first leaf, plant height, number of primary branches, and pod curvature. At both locations, seed yield was positively correlated with the size of the prophyll and negatively correlated with the number of primary branches.

## Association analysis

We found 225 and 213 SNPs, above $-\log10(P) = 3$, associated with the measured traits at Wadi and Giza, respectively (Fig 2 and S1 Table).

Twenty-six SNPs were identified with $-\log10 (P) > 4$ (Table 2), of which 18 SNP were significant at Wadi. One SNP, dDocent_Contig_48195_66, associated with the size of the prophyll at Wadi and Giza, explained 0.21 and 0.23 of the variance, respectively. The allele A of this SNP decreased the size of the prophyll at both locations. The alternate allele for this SNP was G (Table 2).

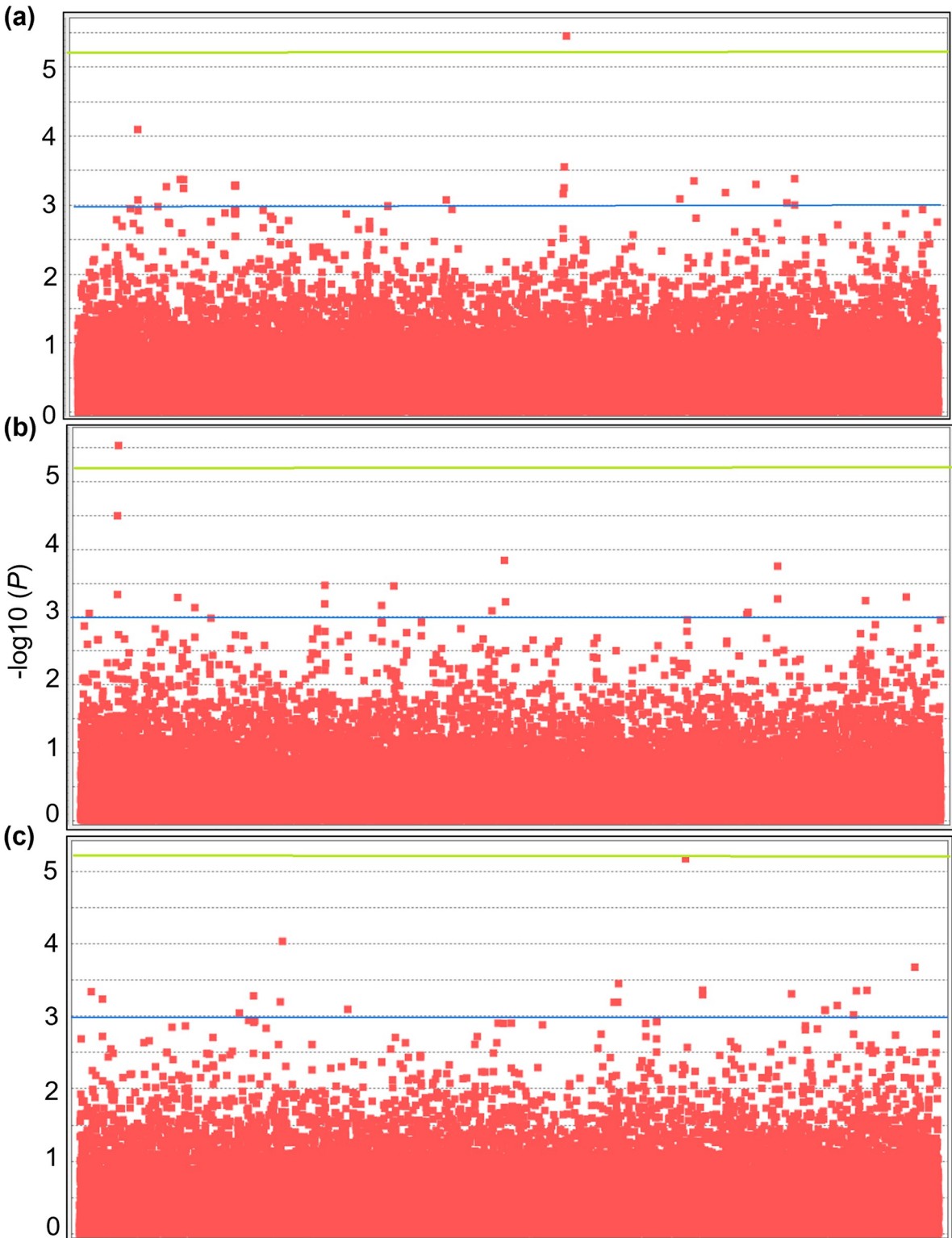

**Fig 2. Manhattan plots for the measured traits.** Significant SNPs associated with the measured traits for the 112 fenugreek genotypes using 38,142 SNPs arranged randomly on the x-axis. The y-axis represents the–log10 (*P*) values. Thresholds defined by a green horizontal line at 5.2 represent FDR, and a blue horizontal line represents an arbitrary threshold = 3. Manhattan plots represent the size of the prophyll measured at Giza (a), the apex shape of the cotyledonary leaf measured at Wadi (b), and the size of the prophyll measured at Wadi (c).

**Table 2. Significant SNPs with–log10 ($P$) $\geq$ 4.**

|  | Trait | Marker | -LOG10 (P) | R2 | Allele | effect |
|---|---|---|---|---|---|---|
| Wadi | Apex shape of the cotyledonary leaf | dDocent_Contig_7337_163 | 5.53 | 0.23 | A (T) | -0.94 |
|  |  | dDocent_Contig_7196_38 | 4.49 | 0.18 | C (T) | -0.93 |
|  | The basal shape of the cotyledonary leaf | dDocent_Contig_38747_133 | 4.44 | 0.18 | A (C) | 0.06 |
|  |  | dDocent_Contig_47861_123 | 4.60 | 0.18 | A (C) | -0.13 |
|  |  | dDocent_Contig_47861_133 | 4.32 | 0.17 | A (G) | -0.09 |
|  |  | dDocent_Contig_47861_23 | 4.32 | 0.17 | G (T) | 0.09 |
|  |  | dDocent_Contig_47861_40 | 4.60 | 0.18 | A (G) | -0.13 |
|  |  | dDocent_Contig_47861_72 | 4.02 | 0.16 | C (T) | 0.06 |
|  | Apex shape of the prophyll | dDocent_Contig_4118_227 | 4.40 | 0.17 | C (T) | -0.86 |
|  |  | dDocent_Contig_4118_186 | 4.03 | 0.16 | A (C) | 0.81 |
|  | The basal shape of the prophyll | dDocent_Contig_11164_213 | 4.13 | 0.16 | A (C) | 0.43 |
|  |  | dDocent_Contig_30941_254 | 4.21 | 0.16 | A (G) | 0.43 |
|  |  | dDocent_Contig_38338_292 | 4.46 | 0.18 | G (T) | 0.75 |
|  |  | dDocent_Contig_39624_229 | 4.26 | 0.20 | C (T) | -0.37 |
|  | Size of the prophyll | dDocent_Contig_48195_66 | 5.19 | 0.21 | A (G) | -1.86 |
|  | The apex shape of the first leaf | dDocent_Contig_2118_276 | 4.18 | 0.16 | C (T) | -1.09 |
|  | The basal shape of the first leaf | dDocent_Contig_43514_115 | 4.16 | 0.16 | G (T) | 0.81 |
|  | Number of primary branches | dDocent_Contig_66850_138 | 4.77 | 0.19 | A (G) | -0.87 |
| Giza | Apex shape of the prophyll | dDocent_Contig_4338_188 | 4.45 | 0.18 | G (T) | 0.78 |
|  | The basal shape of the prophyll | dDocent_Contig_6306_100 | 4.25 | 0.17 | C (T) | -0.55 |
|  |  | dDocent_Contig_7798_186 | 4.34 | 0.17 | A (G) | -0.68 |
|  | Size of the prophyll | dDocent_Contig_10763_122 | 4.10 | 0.16 | A (G) | 1.67 |
|  |  | dDocent_Contig_48195_66 | 5.45 | 0.23 | A (G) | -2.11 |
|  | The apex shape of the first leaf | dDocent_Contig_74819_76 | 4.16 | 0.23 | C (G) | -1.19 |
|  |  | dDocent_Contig_74819_90 | 4.19 | 0.23 | C (G) | -1.19 |
|  | Size of the first leaf | dDocent_Contig_30562_65 | 4.57 | 0.18 | A (G) | 0.63 |
|  | Seed yield |  |  |  |  |  |

SNPs were detected using a mixed linear model and associated with the measured traits at the Wadi and Giza locations. R2 = explained phenotypic variance. The effect of each allele (alternate allele). The 50 nucleotide sequences flanking each SNP are presented in the S1 Table.

Additionally, 30 SNPs were associated with the same traits over the two locations (Table 3). The same allele of each of these SNPs showed the same effect on the associated trait at both locations. Of the 30 SNPs, 18 increased, and 12 decreased the associated traits. Twenty and 14 SNPs were associated with SY measured at Wadi and Giza, respectively.

## Discussion

Breeders invest a lot of money and an extended time, often 10 to 15 years, to develop a new plant variety. Therefore, their efforts should be protected by special rights based on DUS test-ing in nations and territories ratifying the UPOV system. Failing to pass the DUS process entails losing the time and resources spent establishing a variety [23].

The present work sought to morphologically dissect a collection of Egyptian fenugreek plants based on DUS traits across two locations. Using numerical taxonomy to determine cul-tivar distinctness or similarity must inevitably represent a limited data set; otherwise, it becomes impossible to collect every conceivable piece of information [24]. Similar to agro-nomic performance data, many DUS traits are influenced by the environment, undermining

**Table 3. Overlapping significant SNPs between Wadi and Giza.**

| Trait | SNP | Wadi | | | Giza | | |
|---|---|---|---|---|---|---|---|
| | | Allele | Effect | -LOG 10(P) | Allele | Effect | -LOG 10(P) |
| Size of the prophyll | dDocent_Contig_10769_87 | C(T) | 1.8 | 3.25 | C(T) | 2.0 | 3.08 |
| | dDocent_Contig_29296_106 | A(C) | -2.0 | 3.74 | A(C) | -1.2 | 3.27 |
| | dDocent_Contig_29917_12 | C(T) | 1.3 | 3.01 | C(T) | 1.5 | 3.24 |
| | dDocent_Contig_29917_34 | C(T) | -1.3 | 3.01 | C(T) | -1.5 | 3.24 |
| | dDocent_Contig_29917_92 | C(T) | -1.4 | 3.23 | C(T) | -1.6 | 3.37 |
| | dDocent_Contig_41194_183 | C(T) | 1.3 | 3.07 | C(T) | 1.5 | 3.08 |
| | dDocent_Contig_47950_51 | C(T) | -1.3 | 3.03 | C(T) | -1.4 | 3.16 |
| | dDocent_Contig_48195_66 | A(G) | -1.9 | 5.19 | A(G) | -2.1 | 5.45 |
| | dDocent_Contig_66011_261 | A(G) | -1.5 | 3.29 | A(G) | -1.7 | 3.30 |
| | dDocent_Contig_73562_208 | A(G) | 1.6 | 3.30 | A(G) | 1.8 | 3.38 |
| The apex shape of the first leaf | dDocent_Contig_10388_201 | A(G) | 0.7 | 3.34 | A(G) | 0.7 | 3.06 |
| | dDocent_Contig_11143_359 | C(T) | -0.7 | 3.49 | C(T) | -0.7 | 3.19 |
| | dDocent_Contig_2118_276 | C(T) | -1.1 | 4.18 | C(T) | -1.1 | 3.92 |
| | dDocent_Contig_2118_321 | A(G) | -1.2 | 3.89 | A(G) | -1.2 | 3.65 |
| | dDocent_Contig_41601_147 | G(T) | -0.7 | 3.49 | G(T) | -0.7 | 3.19 |
| | dDocent_Contig_43129_261 | C(T) | 0.6 | 3.96 | C(T) | 0.6 | 3.56 |
| | dDocent_Contig_47889_312 | C(T) | 0.5 | 3.05 | C(T) | 0.5 | 3.26 |
| | dDocent_Contig_82708_208 | G(C) | -0.7 | 3.76 | G(C) | -0.6 | 3.35 |
| The basal shape of the first leaf | dDocent_Contig_31097_10 | A(G) | -1.0 | 3.20 | A(G) | -0.9 | 3.21 |
| | dDocent_Contig_31097_130 | A(C) | 1.0 | 3.20 | A(C) | 0.9 | 3.21 |
| | dDocent_Contig_31097_16 | G(T) | 1.0 | 3.20 | G(T) | 0.9 | 3.21 |
| | dDocent_Contig_36120_144 | A(T) | 0.7 | 3.51 | A(T) | 0.8 | 3.99 |
| | dDocent_Contig_43514_115 | G(T) | 0.8 | 4.16 | G(T) | 0.7 | 3.64 |
| | dDocent_Contig_53342_76 | C(T) | 0.9 | 3.13 | C(T) | 1.0 | 3.94 |
| | dDocent_Contig_55619_163 | A(G) | 0.8 | 3.04 | A(G) | 0.9 | 3.41 |
| | dDocent_Contig_938_381 | G(T) | 0.7 | 3.31 | G(T) | 0.7 | 3.19 |
| Pod curvature | dDocent_Contig_41270_146 | C(G) | 0.7 | 3.08 | C(G) | 0.7 | 3.07 |
| | dDocent_Contig_42357_200 | A(C) | 0.8 | 3.07 | A(C) | 0.9 | 3.69 |
| | dDocent_Contig_59493_255 | A(G) | 0.9 | 3.36 | A(G) | 0.9 | 3.16 |
| | dDocent_Contig_78426_62 | A(G) | 0.7 | 3.12 | A(G) | 0.7 | 3.23 |

SNPs were detected using a mixed linear model with–log10 ($P$) $\geq$ 3, and associated with the measured traits. The effect of each allele (alternate allele). The 50 nucleotide sequences flanking each SNP are presented in the S1 Table.

their suitability for taxonomic studies [25, 26]. This observation was similar to our results, where six morphological DUS traits were not correlated, indicating they were not replicable outside the DUS trial. Hence, these traits have limited meaning to variety fingerprinting [15]. In contrast, six other traits showed significant positive correlations between the same trait over the two locations, indicating a higher level of stability and perhaps even a potential role in the emergence of new varieties.

Due to the inconsistencies of the current DUS system across environments, limitations in trait combinatorial space, and impairment caused by a non-optimal marker system, the concept of genomic DUS was introduced to address various issues in the current DUS system [15]. Molecular markers have proven their efficiency in identifying and differentiating varieties regardless of environmental factors. As a result, studies demonstrating the utility of molecular markers in DUS testing continue to pique the interest of plant breeders and UPOV testing

institutions worldwide [23]. Therefore, the UPOV approved the application of molecular markers on the DUS test under three models. However, the potential application of the first UPOV model in fenugreek requires the development of molecular markers associated with genes underlying various agronomic and non-agronomical traits [27], which is impractical due to the limited information on the fenugreek genome [20]. Regarding the third UPOV model, there is also a big argument because the determination of distinctness at three SNPs differences can lead to an inaccurate conclusion of the uniformity and stability results [27]. Previous studies on the second UPOV model found that the correlation between the molecular and phenotypic distance is based on the type, the number, and the distribution of molecular markers and enough phenotypic data with common environmental effects [27, 28].

As a trial to apply the second UPOV model to our data, we used the fenugreek population, genotyped with 28,142 polymorphic SNPs using ddRAD-seq technology [20]. A common disadvantage of the association study approach is the false positive association due to the population structure [17, 29]. Therefore, two statistical correction approaches were validated to set a threshold value at $\alpha = 0.05$ [30] to help overcome this issue. Bonferroni correction [31], as the first approach, calculates the significant $P$-value as the ratio between the significant threshold ($\alpha$) and the number of markers (n). In the second approach, known as FDR [32], $P$-values are arranged in ascending order, then dividing the rank of the p-value (r) by the number of markers (n): ((r/n)×$\alpha$). We detected only three significant SNPs above the -log10 ($P$) values = 6.5 and 5.2, calculated based on the stringent Bonferroni correction and FDR, respectively, which is a common observation working with traits likely to be genetically affected by many small-effect QTLs. As correction approaches assume independence of markers, which is not the case, several earlier studies used arbitrary thresholds between three and four [33–38]. Therefore, we report candidate SNPs with thresholds above three, provided they have the same effects on the trait at both locations.

We looked for the SNPs associated with the size of the prophyll, apex, and basal shape of the first leaf and pod curvature, which showed a correlation of at least 0.9 between both locations, indicating their potential suitability for variety fingerprinting. We identified 30 SNPs associated with those traits, and each associated SNP showed similar effects in both locations per trait. Such SNPs with the main effects are significant in breeding programs and can be further used for marker-assisted breeding [19]. Plant height and the number of primary branches were less stable, with a correlation of 0.6 between the two locations, and the associated SNPs are specific to a single environment, conditionally neutral. Therefore, their application in breeding programs is limited [19].

The high heritability observed for the SY at the two locations suggests a high potential for selection and improvement under Egyptian conditions. Generally, if a ''trait's heritability is moderate or high, the underlying genetic factors control the variation, increasing the possibility of mapping the causal genes underlying this variation [39]. In Giza, a potential association was observed between dDocent_Contig_39611_97 with–log10 ($P$) = 4.2 and SY. This SNP increased SY with 56 g/plant and explained 16% of the yield variation. Although this SNP is specific to the Giza environment, it is a good candidate for breeding in similar environments. Another environmental-specific SNP, dDocent_Contig_39939_66, decreased SY in Wadi, an unfavourable trait in breeding programs.

Some agronomic practices, such as minimal irrigation and weed control, maximize fenugreek yields [40, 41]. Unlike heavy and wet soils, well-drained loamy or sandy soils are ideal for growing fenugreek [5, 9, 42]. When grown in rotation with other annual crops without herbicides, weeds contributed 37–86% of the total fenugreek dry matter yield, whereas controlling annual weeds resulted in no yield loss [41]. In contrast to Giza, the sandy soil and the dripping irrigation system in Wadi explain the higher yield obtained in Wadi.

Several studies on other plants illustrated that the maternal habitat [43–46] and seed priming [47–49] influence offspring adaptive plasticity to tolerate abiotic stresses. Furthermore, previous research on fenugreek reported its ability to withstand a wide range of sodicity at various growth stages [50–53]. Altogether, they illustrate the power of our population to grow in the newly reclaimed saline sandy soil irrigated with saline water in Wadi, as most of the genotypes were collected from Upper Egypt, Alexandria, Faiyum, and Ismailia governorates with sandy soil of different degrees of salinity.

The inconsistency observed here for some DUS traits highlights the significance of molecular markers in the so-called genomic DUS. The same applies to the SY; however, based on SY performance, the selected superior genotypes are good candidates for introgressing the desired genes through breeding programs. Our findings demonstrated "fenugreek's potential to withstand salt stress. More association studies on the morphological, physiological, agronomical, and biochemical levels is required in fenugreek to unravel its ability to tolerate salinity and other abiotic and biotic stresses.

## Supporting information

**S1 Table. Significant SNPs associated with the measured traits at Wadi and Giza with their explained variances and effects.** The 50 nucleotide sequences flanking each SNP are presented in the S1 Table.
(XLSX)

## Acknowledgments

The authors express their great appreciation to Professor Ahmed Medhat Al-Naggar, Department of Agronomy, Faculty of Agriculture, Cairo University, for his comments on the draft of this manuscript.

## Author Contributions

**Conceptualization:** Mustafa M. H. Abd El-Wahab.

**Data curation:** Mohamed El-Soda.

**Formal analysis:** Hashim Abdel-Lattif, Mohamed El-Soda.

**Investigation:** Mustafa M. H. Abd El-Wahab, Kh. S. Emara.

**Project administration:** Mustafa M. H. Abd El-Wahab.

**Resources:** Maha Aljabri.

**Writing – original draft:** Hashim Abdel-Lattif, Kh. S. Emara, Mohamed Mosalam, Mohamed El-Soda.

**Writing – review & editing:** Mohamed El-Soda.

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
