## [Decision Letter · Decision Letter 0]

15 Aug 2023

PONE-D-23-24344Mapping SNP markers associated with distinctness, uniformity, and stability testing in Egyptian fenugreek genotypesPLOS ONE

Dear Dr. El-Soda,

Thank you for submitting your manuscript to PLOS ONE. After careful consideration, we feel that it has merit but does not fully meet PLOS ONE’s publication criteria as it currently stands. Therefore, we invite you to submit a revised version of the manuscript that addresses the points raised during the review process.

We look forward to receiving your revised manuscript.

Kind regards,

Andrea Mastinu

Academic Editor

PLOS ONE

Journal Requirements:

Reviewers' comments:

Reviewer's Responses to Questions

**Comments to the Author**

1. Is the manuscript technically sound, and do the data support the conclusions?

Reviewer #1: Partly

2. Has the statistical analysis been performed appropriately and rigorously? 

Reviewer #1: Yes

3. Have the authors made all data underlying the findings in their manuscript fully available?

Reviewer #1: No

4. Is the manuscript presented in an intelligible fashion and written in standard English?

Reviewer #1: Yes

5. Review Comments to the Author

Reviewer #1: The manuscript PONE-D-23-24344 deals with very interesting and important topic for SNP identification associated with traits for distinctness, uniformity, and stability (DUS) in Egyptian fenugreek. However, about 1/3 part of the manuscript was missed and, therefore, in current form, the manuscript is absolutely unacceptable for evaluation. Authors must address firstly Major points and following Minor points indicated below. The fully revised manuscript, where all points are addressed, can be re-considered again.

Major comments and suggestions:

(1) The manuscript is about identified SNP and developed markers starting from Title and through the entire manuscript. However, there is no description of what, where and how SNP were developed at all. The sub-section about SNP and their identification is missed completely from M&M section, and L180-186 is already about Association mapping. In Abstract (L34-35), authors wrote: “…we performed an association mapping, and out of 38,142 SNPs, we identified…”. Therefore, I can suppose that authors somewhere developed such large amount of SNP. In Legend of Figure 2 (L236), authors again wrote: “…using 38,142 SNPs arranged randomly …”. This indicate that there is no mistake and authors have somewhere these SNPs, but this is unclear why they did not describe it at all in M&M section? The ‘shade of mystery’ was started to disappear in the Discussion, where in L298-300, authors disclosed as follows: “…we used the fenugreek population, genotyped with 28,142 polymorphic SNPs using double digest restriction site-associated DNA sequencing (ddRAD-seq) technology [21]”. I suppose the reducing for 10,000 SNP is accidental mistake because I found publication [21] of the same authors in journal ‘Genes’ (MDPI) in 2020, and it was mentioned for 38,142 but not 28,142 polymorphic SNP. Finally, this story becomes more clear when I have read the published paper [21] in ‘Genes’. In fact, the current manuscript is a direct continuation of authors’ previous study because 38,142 SNPs were perfectly described and experiments were carried out with the same 112 fenugreek genotypes (L126 and L235-236) but not 107 genotypes as accidently written in Abstract (L27). Therefore, authors must read their own published paper [21] (Genes, 2020) and write in M&M sub-section about SNP and ddRAD-seq with their reference to published paper [21]. However, this is not enough and authors must describe in the current manuscript briefly but clear and sufficient, how 38,142 polymorphic SNP were identified and how they were developed using ddRAD-seq technology? Moreover, sequences of ‘dDocent-Contigs’ listed in Tables 2 and 3 MUST be provided in Supplementary material, similar to those as it was published in [21]. Additionally, sequences of ‘dDocent-Contigs’ and three SNPs identified and shown in Figure 2 must be present in the additional Table in the main text body of the manuscript, similar to those in [21]. Only after that, this is possible to re-read and re-evaluate the revised manuscript with clear description.

(2) The Title is started from the term ‘mapping’ and it constantly occurs again and again in the text, for example, ‘we mapped’ (L232) and ‘SNPs were mapped’ (L242). Additionally, authors used ‘association mapping’ method (L180, L230, L301, L309, L343). However, there is no ‘mapping’ in this manuscript. Authors mentioned that they used ‘Association mapping’ method but this method is based on well-known SNP and their localization on genetic or physical map of fenugreek. Below, I provide a citation of an article ‘Association mapping’ from Wikipedia:

“The advantage of association mapping is that it can map quantitative traits with high resolution in a way that is statistically very powerful. Association mapping, however, also requires extensive knowledge of SNPs within the genome of the organism of interest, and is therefore difficult to perform in species that have not been well studied or do not have well-annotated genomes”.

In fact, authors used one of the options of ‘Association mapping’ method, which is closer to ‘Genome-wide association study’ (GWAS). Both methods can use molecular markers with established genetic map and known localization of the markers in linkage groups. However, GWAS is based just on association between phenotypic traits and molecular markers regardless their known or unknown localization on chromosomes. This is exactly happened and present in the current manuscript, Figure 2 with Manhattan plots, where 38,142 SNPs arranged randomly on the x-axis (L236). Why ‘randomly’? This is because their genetic mapping remains unknown.

Despite published earlier paper [21] by the same authors in 2020 with ‘association mapping’ in the Title, I strongly ask authors to revise their current manuscript and use ‘association study’ and remove ‘association mapping’. Additionally, authors have to present a paragraph in Discussion section with their statement explaining the situation around ‘association study’ and ‘association mapping’. This is very important for readers to understand how authors manage with this issue.

Minor notes and corrections:

(3) L27 and L126. Please correct, how many fenugreek accessions were used.

(4) L31. Please replace the abbreviation ‘SY’; and use the full term ‘seed yield’ in the Abstract.

(5) L38-40, L310-312, L339-346 and in other parts. Authors have to be careful and do not mix two items: (A) Stability and variation of DUS-related traits; and (B) MAS. Because DUS-related traits are in the main focus of the manuscript, starting from the Title, authors have to discuss it everywhere first. The second item (MAS) is related to the improvement of existing genotypes. Therefore, authors have to revise their manuscript and collect all their statements about MAS in separate part in the end of Discussion only. Similar, in Abstract, author can mention MAS as potential application of the presented study in future.

(6) L184-185. I cannot understand why authors selected thresholds above 3, if FDR was 5.9? Moreover, threshold was variable: above 5 in Figure 2 (L235), above 4 in Table 2 (L246) and above 3 in Table 3 (L257). Please explain and insert a summary of this explanation in the text (M&M or Discussion).

(7) L234-239 and Figure 2. The Figure with legend is unacceptable in the current form. Please your previous paper [21], where Figure 4 was present in much better style. You must modify Figure 2 accordingly: insert dash lines for threshold levels, names of contigs for SNP above the thresholds with proper description in the Figure legend.

(8) L245. If authors declare one allele is associated with decreased size of the prophyll, please complete the statement and add which allele is association with bigger size of seeds.

6. PLOS authors have the option to publish the peer review history of their article (what does this mean?). If published, this will include your full peer review and any attached files.

Reviewer #1: **Yes: **Yuri Shavrukov

---

## [Author Response · Author response to Decision Letter 0]

30 Aug 2023

5. Review Comments to the Author

Reviewer #1: The manuscript PONE-D-23-24344 deals with very interesting and important topic for SNP identification associated with traits for distinctness, uniformity, and stability (DUS) in Egyptian fenugreek. However, about 1/3 part of the manuscript was missed and, therefore, in current form, the manuscript is absolutely unacceptable for evaluation. Authors must address firstly Major points and following Minor points indicated below. The fully revised manuscript, where all points are addressed, can be re-considered again.

We want to thank the esteemed reviewer for his constructive comments. 

Major comments and suggestions:

(1) The manuscript is about identified SNP and developed markers starting from Title and through the entire manuscript. However, there is no description of what, where and how SNP were developed at all. The sub-section about SNP and their identification is missed completely from M&M section, and L180-186 is already about Association mapping. In Abstract (L34-35), authors wrote: "…we performed an association mapping, and out of 38,142 SNPs, we identified…". Therefore, I can suppose that authors somewhere developed such large amount of SNP. In Legend of Figure 2 (L236), authors again wrote: "…using 38,142 SNPs arranged randomly …". This indicate that there is no mistake and authors have somewhere these SNPs, but this is unclear why they did not describe it at all in M&M section? The 'shade of mystery' was started to disappear in the Discussion, where in L298-300, authors disclosed as follows: "…we used the fenugreek population, genotyped with 28,142 polymorphic SNPs using double digest restriction site-associated DNA sequencing (ddRAD-seq) technology [21]". I suppose the reducing for 10,000 SNP is accidental mistake because I found publication [21] of the same authors in journal 'Genes' (MDPI) in 2020, and it was mentioned for 38,142 but not 28,142 polymorphic SNP. Finally, this story becomes more clear when I have read the published paper [21] in 'Genes'. In fact, the current manuscript is a direct continuation of authors' previous study because 38,142 SNPs were perfectly described and experiments were carried out with the same 112 fenugreek genotypes (L126 and L235-236) but not 107 genotypes as accidently written in Abstract (L27). Therefore, authors must read their own published paper [21] (Genes, 2020) and write in M&M sub-section about SNP and ddRAD-seq with their reference to published paper [21]. However, this is not enough and authors must describe in the current manuscript briefly but clear and sufficient, how 38,142 polymorphic SNP were identified and how they were developed using ddRAD-seq technology? Moreover, sequences of 'dDocent-Contigs' listed in Tables 2 and 3 MUST be provided in Supplementary material, similar to those as it was published in [21]. Additionally, sequences of 'dDocent-Contigs' and three SNPs identified and shown in Figure 2 must be present in the additional Table in the main text body of the manuscript, similar to those in [21]. Only after that, this is possible to re-read and re-evaluate the revised manuscript with clear description.

- Thanks for the comments, and I apologise for the typo error in the Abstract. The number of SNPs and genotypes has been corrected. 

- We are very thankful to the reviewer whose comments improved the manuscript. A paragraph describing the genotypic data has been added, lines 183 – 187.

- The sequences of all 'dDocent-Contigs' identified here were added to the supplementary data 1, as it contains all SNPs in Tables 2 and 3. Adding sequences to tables 2 and 3 would enlarge the size of both tables. This information was added to lines 256, 267, and 481. 

(2) The Title is started from the term 'mapping' and it constantly occurs again and again in the text, for example, 'we mapped' (L232) and 'SNPs were mapped' (L242). Additionally, authors used 'association mapping' method (L180, L230, L301, L309, L343). However, there is no 'mapping' in this manuscript. Authors mentioned that they used 'Association mapping' method but this method is based on well-known SNP and their localization on genetic or physical map of fenugreek. Below, I provide a citation of an article 'Association mapping' from Wikipedia:

"The advantage of association mapping is that it can map quantitative traits with high resolution in a way that is statistically very powerful. Association mapping, however, also requires extensive knowledge of SNPs within the genome of the organism of interest, and is therefore difficult to perform in species that have not been well studied or do not have well-annotated genomes".

In fact, authors used one of the options of 'Association mapping' method, which is closer to 'Genome-wide association study' (GWAS). Both methods can use molecular markers with established genetic map and known localization of the markers in linkage groups. However, GWAS is based just on association between phenotypic traits and molecular markers regardless their known or unknown localization on chromosomes. This is exactly happened and present in the current manuscript, Figure 2 with Manhattan plots, where 38,142 SNPs arranged randomly on the x-axis (L236). Why 'randomly'? This is because their genetic mapping remains unknown.

Despite published earlier paper [21] by the same authors in 2020 with 'association mapping' in the Title, I strongly ask authors to revise their current manuscript and use' association study' and remove 'association mapping'. Additionally, authors have to present a paragraph in Discussion section with their statement explaining the situation around 'association study' and 'association mapping'. This is very important for readers to understand how authors manage with this issue.

Thank you for the suggestion to change "mapping" to "study". We have done that throughout the manuscript. However, I don't see the point of discussing "the situation around association study and 'association mapping". In fact, both terms are used interchangeably. For example, the article "Genome-Wide Association Mapping of Flowering and Ripening Periods in Apple" published in Frontiers in Plant Sciences in 2017, the article "Conditions Under Which Genome-Wide Association Studies Will be Positively Misleading" published in Genetics in 2010, and several more other articles are using both terms interchangeably between the title and the body text. 

Another point is that, although we have changed mapping to study, as per the reviewer's suggestion, it is well known that hundreds of earlier studies used the word "mapping" without available knowledge of markers' positions. For example, several earlier QTL mapping studies used "linkage groups" instead of chromosomes because no sequences were known. In addition, and in many cases, the same marker could be mistakenly mapped to different chromosomes, which is well-known and observed in many plants. 

Therefore, the paragraph discussing "the situation around association study and association mapping" was not added. 

Minor notes and corrections:

(3) L27 and L126. Please correct, how many fenugreek accessions were used.

Thank you very much. Corrected in line 27. 

(4) L31. Please replace the abbreviation 'SY'; and use the full term' seed yield' in the Abstract.

Thank you very much. Corrected. 

(5) L38-40, L310-312, L339-346 and in other parts. Authors have to be careful and do not mix two items: (A) Stability and variation of DUS-related traits; and (B) MAS. Because DUS-related traits are in the main focus of the manuscript, starting from the Title, authors have to discuss it everywhere first. The second item (MAS) is related to the improvement of existing genotypes. Therefore, authors have to revise their manuscript and collect all their statements about MAS in separate part in the end of Discussion only. Similar, in Abstract, author can mention MAS as potential application of the presented study in future.

Thank you very much for this valuable comment. We have restructured the Discussion as advised. 

(6) L184-185. I cannot understand why authors selected thresholds above 3, if FDR was 5.9? Moreover, threshold was variable: above 5 in Figure 2 (L235), above 4 in Table 2 (L246) and above 3 in Table 3 (L257). Please explain and insert a summary of this explanation in the text (M&M or Discussion).

Thank you very much for this valuable comment. We discussed this issue in lines 331 – 344.

(7) L234-239 and Figure 2. The Figure with legend is unacceptable in the current form. Please your previous paper [21], where Figure 4 was present in much better style. You must modify Figure 2 accordingly: insert dash lines for threshold levels, names of contigs for SNP above the thresholds with proper description in the Figure legend.

We have added 2 lines to present the two thresholds, based on FDR = 5.2 (green) and an arbitrary threshold 3 (blue). We changed the legends, lines 242 – 248. Adding each contig's name was impossible because the Figure would be filled and not readable. 

(8) L245. If authors declare one allele is associated with decreased size of the prophyll, please complete the statement and add which allele is association with bigger size of seeds.

We added this to line 255.

---

## [Decision Letter · Decision Letter 1]

31 Aug 2023

Identifying SNP markers associated with distinctness, uniformity, and stability testing in Egyptian fenugreek genotypes

PONE-D-23-24344R1

Dear Dr. El-Soda,

We’re pleased to inform you that your manuscript has been judged scientifically suitable for publication and will be formally accepted for publication once it meets all outstanding technical requirements.

Kind regards,

Andrea Mastinu

Academic Editor

PLOS ONE

Additional Editor Comments (optional):

Reviewers' comments:

Reviewer's Responses to Questions

**Comments to the Author**

1. If the authors have adequately addressed your comments raised in a previous round of review and you feel that this manuscript is now acceptable for publication, you may indicate that here to bypass the “Comments to the Author” section, enter your conflict of interest statement in the “Confidential to Editor” section, and submit your "Accept" recommendation.

Reviewer #1: All comments have been addressed

2. Is the manuscript technically sound, and do the data support the conclusions?

Reviewer #1: Yes

3. Has the statistical analysis been performed appropriately and rigorously? 

Reviewer #1: Yes

4. Have the authors made all data underlying the findings in their manuscript fully available?

Reviewer #1: Yes

5. Is the manuscript presented in an intelligible fashion and written in standard English?

Reviewer #1: Yes

6. Review Comments to the Author

Reviewer #1: Authors made their great job and addressed all comments properly. I have no further comment and wish all the best to authors.

7. PLOS authors have the option to publish the peer review history of their article (what does this mean?). If published, this will include your full peer review and any attached files.

Reviewer #1: **Yes: **Yuri Shavrukov

---

## [Editor Report · Acceptance letter]

12 Sep 2023

PONE-D-23-24344R1 

Identifying SNP markers associated with distinctness, uniformity, and stability testing in Egyptian fenugreek genotypes 

Dear Dr. El-Soda:

I'm pleased to inform you that your manuscript has been deemed suitable for publication in PLOS ONE. Congratulations! Your manuscript is now with our production department. 

Kind regards, 

on behalf of

Dr. Andrea Mastinu 

Academic Editor

PLOS ONE